# Real-World Implementation of Precision Psychiatry: A Systematic Review of Barriers and Facilitators

**DOI:** 10.3390/brainsci12070934

**Published:** 2022-07-16

**Authors:** Helen Baldwin, Lion Loebel-Davidsohn, Dominic Oliver, Gonzalo Salazar de Pablo, Daniel Stahl, Heleen Riper, Paolo Fusar-Poli

**Affiliations:** 1Early Psychosis: Interventions and Clinical-Detection (EPIC) Lab, Department of Psychosis Studies, Institute of Psychiatry, Psychology & Neuroscience, King’s College London, 16 De Crespigny Park, London SE5 8AF, UK; dominic.a.oliver@kcl.ac.uk (D.O.); gonzalo.salazar_de_pablo@kcl.ac.uk (G.S.d.P.); paolo.fusar-poli@kcl.ac.uk (P.F.-P.); 2National Institute for Health Research, Maudsley Biomedical Research Centre, South London and Maudsley National Health Service Foundation Trust, London SE5 8AF, UK; 3Department of Molecular Medicine, Faculty of Medicine and Surgery, University of Pavia, 27100 Pavia, Italy; lion.loebeldavidsoh01@universitadipavia.it; 4Institute of Psychiatry and Mental Health, Department of Child and Adolescent Psychiatry, Hospital General Universitario Gregorio Marañón School of Medicine, Universidad Complutense, Instituto de Investigación Sanitaria Gregorio Marañón, CIBERSAM, 28009 Madrid, Spain; 5Departent of Child and Adolescent Psychiatry, Institute of Psychiatry, Psychology & Neuroscience, King’s College London, London SE5 8AF, UK; 6Child and Adolescent Mental Health Services, South London and Maudsley National Health Service Foundation Trust, London SE5 8AZ, UK; 7Biostatistics Department, Institute of Psychiatry, Psychology and Neuroscience, King’s College London, London SE5 8AF, UK; daniel.r.stahl@kcl.ac.uk; 8Department of Clinical, Neuro and Developmental Psychology, VU University, 1081 HV Amsterdam, The Netherlands; h.riper@vu.nl; 9Department of Psychiatry, Amsterdam University Medical Centre (VUmc), 1081 HV Amsterdam, The Netherlands; 10Department of Psychiatry, Faculty of Medicine, University of Turku, 20700 Turku, Finland; 11OASIS Service, South London and Maudsley National Health Service Foundation Trust, London SE11 5DL, UK; 12Department of Brain and Behavioural Sciences, University of Pavia, 27100 Pavia, Italy

**Keywords:** precision medicine, psychiatry, real-world implementation, systematic review, barriers, facilitators

## Abstract

Background: Despite significant research progress surrounding precision medicine in psychiatry, there has been little tangible impact upon real-world clinical care. Objective: To identify barriers and facilitators affecting the real-world implementation of precision psychiatry. Method: A PRISMA-compliant systematic literature search of primary research studies, conducted in the Web of Science, Cochrane Central Register of Controlled Trials, PsycINFO and OpenGrey databases. We included a qualitative data synthesis structured according to the ‘Consolidated Framework for Implementation Research’ (CFIR) key constructs. Results: Of 93,886 records screened, 28 studies were suitable for inclusion. The included studies reported 38 barriers and facilitators attributed to the CFIR constructs. Commonly reported barriers included: potential psychological harm to the service user (*n* = 11), cost and time investments (*n* = 9), potential economic and occupational harm to the service user (*n* = 8), poor accuracy and utility of the model (*n* = 8), and poor perceived competence in precision medicine amongst staff (*n* = 7). The most highly reported facilitator was the availability of adequate competence and skills training for staff (*n* = 7). Conclusions: Psychiatry faces widespread challenges in the implementation of precision medicine methods. Innovative solutions are required at the level of the individual and the wider system to fulfil the translational gap and impact real-world care.

## 1. Introduction

Precision medicine describes the tailoring of health care to an individual and their unique biological, social, and/or environmental profile, or such tailoring across stratified subgroups [1]. Precision medicine replaces the more outdated term ‘personalized medicine’ due to fears that the term could be misinterpreted to suggest that specific interventions could be developed for each unique individual, which is not the case. Whilst these concepts have existed in somatic medicine for decades, such as the practice of matching the blood type of transfusion patients to the donor, it has been introduced in psychiatry only recently [2]. Furthermore, recent advances in data-driven mental health care have offered further promise of embedding precision medicine more firmly across mental health care services [3,4,5], in the form of diagnostic (the probability that a particular condition is present), prognostic (probability of particular outcomes), and prediction (forecasting the response to specific interventions) models [6]. Indeed, demonstrable progress has already been observed in the delivery of oncology [7] and cardiology [8] services, amongst other fields of medicine. However, psychiatry is yet to observe similar translational success and numerous criticisms have emerged [9,10,11].

Despite significant progress in the development and validation of clinical prediction models across psychiatric research, few of these advances have been successfully implemented into clinical practice. In fact, a recent systematic review reported that less than 1% of individualized prediction models existing in psychiatric literature are considered for actual implementation in real-world care [6]. This clear translational gap necessitates a closer examination of implementation barriers. Whilst there are a range of informational, regulatory and logistical barriers which impede the progress of precision medicine in general [12,13,14,15,16,17], psychiatry may also pose distinct challenges in light of the phenomenological complexity and heterogeneous nature of psychiatric conditions [3,18], the lack of established pathophysiological pathways in psychiatry [3,19], and unique ethical considerations associated with the historically complex socio-political perceptions and attitudes towards mental illness and psychiatry [20,21]. These additional challenges necessitate the systematic identification of key factors that obstruct or promote the implementation of precision psychiatry, to create an empirical framework in order to devise appropriate solutions at the level of the individual, the organization and the wider system.

### Aims

This current review aims to systematically examine the existing literature concerning the key barriers to, and facilitators of, the implementation of precision psychiatry into real-world clinical practice. As such, we aim to develop a framework of barriers and facilitators, structured in accordance with the Consolidated Framework for Implementation Research (CFIR), in order to inform the development of future precision medicine models in psychiatry.

## 2. METHODS

### 2.1. Search Strategy and Selection Criteria

We conducted a PRISMA-compliant (Appendix A: PRISMA 2020 Statement and Checklist) systematic literature review, including a systematic search strategy (Appendix A) conducted in Web of Science (including Web of Science Core Collection, KCI-Korean Journal Database, MEDLINE, Russian Science Citation Index, and SciELO Citation Index), the Cochrane Central Register of Controlled Trials, PsycINFO via Ovid, and the OpenGrey database up to and including publication on 25 October 2020.

The review protocol (PROSPERO: CRD42020182595) aimed to identify relevant literature which reported an assessment of the factors affecting the real-world implementation of precision psychiatry methods, defined as the application of any method encompassing diagnostic, prognostic, or predictive models to estimate risk or outcomes at an individual- (precision) or subgroup-level (stratified) [2]. There were no restrictions in place regarding the types of predictors in use, and the final literature was clustered against predictors previously validated. The full inclusion and exclusion criteria are listed in Table 1. We also considered literature which reported on the perspectives of a variety of key stakeholders; this allowed us to gain a comprehensive and multidisciplinary assessment of barriers and facilitators throughout all stages of implementation. Any implementation studies or other primary research which featured a qualitative, quantitative or mixed-methods examination of barriers and/or facilitators were considered for inclusion. Given the extreme paucity of actual implementation studies,^6^ we also considered studies for inclusion which adopted a more hypothetical consideration of implementation (i.e., primary research which involved stakeholder consultation on the proposed application of aspects of precision psychiatry methods).

Database results were exported into EndNote X9 and screened within the application. The lead reviewer (HB) conducted title and abstract screening on all exported records in line with the inclusion criteria outlined below. A second reviewer independently screened a random 50% sample of the records (LLD) due to the extensive number of initial database results. All abstracts which appeared to meet the inclusion criteria were carried forward to full-text screening; the entire full-text screening process was conducted by two independent researchers (HB, LLD). Any uncertainties over screening or full-text decisions were consulted with the wider review team and eventually with a senior researcher (PFP).

### 2.2. Level of Evidence

Given the expected narrow final literature base and in the interest of inclusivity, we made the decision not to exclude based upon level or quality of evidence. Instead, we adhered to a simple numerical ranking system to indicate the level of evidence: 1 denotes stakeholder consultation, 2 denotes pilot and feasibility studies, and randomized controlled trials (RCT) will be assigned a score of 3 to denote higher grade evidence.

### 2.3. Data Extraction and Analysis

A fit-for-purpose data extraction form was designed for this review and was first trialed on five studies and subsequently adjusted as necessary before proceeding with the remaining studies. Data extraction was independently conducted by two reviewers (HB, LLD) to ensure all relevant information was captured. Appendix A outline the categories of data which were extracted for each record (Appendix A).

The identified barriers and facilitators were systematically synthesized, modelled upon a systematic approach adopted in a recent review addressing the implementation of digital health interventions for psychosis, and bipolar specifically [22]. As such, the data synthesis was guided by the ‘Consolidated Framework for Implementation Research’ (CFIR) [23], which provides an outline of factors commonly associated with the implementation of innovation into clinical practice, corresponding to five key constructs: intervention characteristics, outer setting, inner setting, characteristics of the individuals and the implementation process (described below). Here, we made two minor revisions to these constructs. First, as many precision psychiatry models are not necessarily intervention-based, the ‘Intervention characteristics’ construct is hereafter referred to as ‘Characteristics of the model’. Second, the ‘Characteristics of the individuals’ construct will be divided into separate ‘Staff’ and ‘Service user’ constructs in recognition of their distinct roles in the implementation process. Sub-factors were then grouped and discussed within each construct post-hoc based upon thematic or conceptual similarities raised within the literature.

▪Characteristics of the model: This construct addresses logistical and practical features of the model which may impact upon implementation, as well as more conceptual components of the model and the corresponding strength, accuracy and transparency of the evidence upon which the model is based.▪Inner setting: Whilst the inner and outer setting constructs are closely linked and are considered inter-dependent in many respects, this construct refers largely to features of local infrastructure within which the model will be implemented.▪Outer setting: Closely intertwined with the inner setting, the outer setting largely takes into consideration the wider system and the external organizations who exist outside of the inner setting, and as such this construct typically addresses the economic, social, cultural and political contexts within which the model is being implemented.▪Characteristics of the individuals: This construct considers the individuals involved in the implementation process at the ground-level, including both those involved in the delivery of clinical care and those in receipt of this care. As such, in this current study, we divided this construct into two independent groups of stakeholders due to their unique needs and perspectives: (i) health and social care staff involved in the delivery of care and (ii) service users and their families/caregivers. These constructs address the attitudes, opinions, previous experiences, skills/knowledge, concerns, needs and potential impact of precision psychiatry models on these key stakeholder groups.▪Implementation process: Finally, this process construct relates to factors which may affect the actual procedure and operations of implementation, including uptake of and adherence to the process.

## 3. Results

### 3.1. Literature Search

Following de-duplication, 93,886 records were screened by abstract, of which 407 were carried forward to full-text screening. A final 28 records met all inclusion criteria (Figure 1).

### 3.2. Description of the Included Studies

The included records were published between 1996 and 2020. The final literature base represented global research from various sites; United States (*n* = 15; 53.6%), United Kingdom (*n* = 3; 10.7%), Australia (*n* = 3; 10.7%), Spain (*n* = 2; 7.1%), Canada (*n* = 1; 3.6%), New Zealand (*n* = 1; 3.6%), Singapore (*n* = 1; 3.6%), Demark (*n* = 1; 3.6%), and one study spanning 22 countries across North and South America, Europe, and Asia-Pacific regions.

Just seven (25.0%) of the 28 included records reported barriers and facilitators derived from the actual real-world implementation of precision psychiatry methods; two feasibility studies [24,25], one case example [26], and four qualitative studies employing surveys and/or interviews [27,28,29,30]. The remaining studies (*n* = 21; 75.0%) [31,32,33,34,35,36,37,38,39,40,41,42,43,44,45,46,47,48,49,50,51] were not primarily based on implementation data but rather based upon qualitative stakeholder consultation on the hypothetical implementation of precision psychiatry methodologies. There were no relevant RCTs identified during the screening process.

A diverse range of stakeholders’ opinions were represented within the final literature base and several studies presented the perspectives of more than one type of stakeholder group including (Figure 2): health care professionals, such as psychiatrists, psychologists, nurses, medical students, and general physicians amongst other health professionals (*n* = 23; 82.1%) [24,25,27,28,29,30,32,33,34,35,36,37,38,39,40,41,42,43,44,45,46,50,51], service users (*n* = 9; 32.1%) [25,31,35,39,41,43,46,47,48], caregivers or family members (*n* = 5; 17.9%) [24,35,39,43,47], community members (*n* = 3; 10.7%) [36,48,49], and research scientists (*n* = 2; 7.1%) [26,36]. There were no included studies which consulted policymakers.

Further, the included literature represented a range of specialist fields of psychiatry (Figure 3); general psychiatry (*n* = 14; 50.0%) [27,28,30,33,34,36,37,38,39,40,42,43,44,45], major depression/mood disorders (*n* = 7; 25.0%) [29,31,35,41,48,49,51], bipolar disorder (*n* = 3; 10.7%) [46,47,51], suicidal behaviors (*n* = 2; 7.1%) [26,32], psychosis (*n* = 1; 3.6%) [51], child psychiatry (*n* = 1; 3.6%) [24], alcohol use disorders (*n* = 1; 3.6%) [50], and the clinical high-risk for psychosis (CHR-P) state (*n* = 1; 3.6%) [25].

There was also a diverse range of precision approaches considered within the literature (Figure 4), including: genetic testing (*n* = 12; 42.9%) [35,36,37,40,42,43,45,46,47,49,50,51], pharmacogenomics (*n* = 7; 25.0%) [24,27,28,30,33,38,44], clinical prediction models and risk calculators (*n* = 6; 21.4%) [25,26,29,31,32,48], clinical decision supports (*n* = 2; 7.1%) [38,39], functional brain imaging (*n* = 1; 3.6%) [41], and general Artificial Intelligence (AI)/ Machine Learning (ML) applications (*n* = 1; 3.6%) [34]. This final literature base discussed a wealth of potential barriers and facilitators which covered the range of established CFIR constructs [23]. Table 2 offers a detailed breakdown of the characteristics of the included studies.

### 3.3. Level of Evidence Summary

As no RCT’s were identified during the review process, no records were assigned a higher-grade score of 3. Seven records (25.0%) were assigned a mid-grade quality score of 2, and the remaining 21 (75.0%) records were assigned a lower-grade score of 1.

### 3.4. Factors Affecting the Implementation of Precision Psychiatry

We identified a broad variety of factors which covered each of the key CFIR constructs. Figure 5 provides a high-level visual summary of the identified barriers and facilitators corresponding to each of these constructs and the reported frequencies of each factor are presented in Figure 6 (barriers) and Figure 7 (facilitators).

#### 3.4.1. Characteristics of the Model

Of the included studies, 57.1% (*n* = 16) reported barriers or facilitators relating to particular characteristics of the precision model which may impact upon implementation.

##### Barriers

Cost and time investments

Logistical and practical barriers associated with cost and time investments of the prediction model were the most highly reported barrier within this construct (*n*= 9; 32.1%). They were almost exclusively reported by health care professionals as opposed to any other stakeholder groups. In particular, the potential financial burden and high costs associated with the implementation of the precision model were highly reported as barriers [24,28,33,39,41,50] alongside the need for data to highlight cost-effectiveness before widespread implementation can be considered [39]. Concerns surrounding time investments were also identified, including the use of time-intensive tools within a small consultation window [27,28,29,38,39], slow computational speed of electronic tools [38], and lengthy waiting times for results [24,33,50]—particularly regarding patients who are acutely unwell [27]. Some physicians reported that the use of a risk prediction model also required more effort to implement than standard practice and led to more frequent patient visits, though this was rebutted by other physicians across a series of interviews [29]. Furthermore, clinicians also highlighted the challenging aspects of alert systems embedded into precision medicine tools and emphasized the need to distinguish between alerts which are clinically useful and those which are not in order to avoid alert fatigue and optimize efficient use of time [38].

2.Poor accuracy and utility of the model

A further highly reported thematic group of barriers (*n* = 8; 28.6%) within this construct related to the predictive accuracy and clinical utility of the precision model. These barriers were also largely reported by health care professionals, and included concerns relating to the substantial margin of error in risk prediction [48], the potential for false-negative outcomes [32], and poor predictability and prognostic accuracy beyond current working practice [36,42,49,50]. Furthermore, there were additional challenges raised with respect to the lack of strength of current evidence [39,50], and the corresponding need for published peer-reviewed supportive evidence [28].

3.Poor perceived relative advantage of the model

Challenges surrounding the relative advantage of the precision model were also frequently highlighted (*n* = 5; 17.9%). Several studies reported stakeholder beliefs that the precision model offered no advantage beyond current clinical care and risked the potential of being implemented at the expense of clinical judgement [27], could pose greater general risks compared to current care [34], did not reflect the complexities of clinical consultation, and could not capture the more nuanced and fine-grained factors that are only available through human interaction which might usually influence the clinical decision-making process [39]. In some instances, testing was only deemed to be advantageous for particular subgroups of service users who might benefit most and as such was only reserved for these select groups, such as those with a family psychiatric history [49], poor treatment responders or those with more atypical symptom presentations [38].

4.Poor transparency and complexity of the model

Several studies also reported challenges relating to the technical complexity of the precision model (*n* = 5; 17.9%) and the corresponding difficulty in translating these concepts to a layperson or to a population with reduced cognitive capacity, therefore highlighting the poor adaptability of the tool to perform equally well with a range of recipients. Specific barriers were identified regarding the lack of transparency over the underlying algorithm and which factors statistically contributed most to a high-risk outcome [32], overly medicalized language [39], the numerical output of a tool in conjunction with low numerical literacy across the population [48], difficulties understanding and communicating the concept of risk prediction [29], and subsequent ethical concerns surrounding capacity to consent to treatment decisions [39,42].

5.Lack of clear guidelines

Finally, a lack of clear guidelines to govern use of the precision model was raised by just one study (3.6%) in relation to the usage of pharmacogenomic methods [33], highlighting corresponding guidance documentation as an important aspect of implementation.

##### Facilitators

6.Simplicity and usability of the model

Two studies (7.1%) placed emphasis upon the importance of a precision model which is both simple and quick to use [29,39], with a range of health care professionals highlighting that a time-saving tool is more likely to be used in practice, particularly in light of typically narrow clinical consultation periods.

7.Collaborative usage

In one study (3.6%), health care professionals also called for patient-facing precision tools, stating that they promote collaboration and build trust between the health care professional and the service user [39].

8.Adaptability of the model

In order to overcome issues surrounding model complexity and poor adaptability (discussed above in Section 3.4.1), one prototype development study (3.6%) reported that patient-facing tools require an emphasis upon clear communication and offered pictorial format as a potential solution [48].

9.Stratification over precision

The way in which the outcome of a precision model was delivered was also of importance, with health care professionals favoring stratified outcome prediction (e.g., low-, medium-, and high-risk) rather than individualized, numerical figures in one study (3.6%) [29].

10.Multi-modal models

Precision models utilising multi-modal data were also raised as a potential facilitating factor in one study (3.6%), as clinicians expressed interest in multi-modal models involving the integration of biological data with lifestyle factors to increase accuracy [39].

11.Routinely collected predictors

Finally, one study (3.6%) highlighted the preferred use of factors which are routinely collected in clinical practice as these are better suited to more seamless implementation [25].

#### 3.4.2. Inner Setting

One half of the included literature (*n* = 14; 50%) reported factors relating to features of the inner setting.

##### Barriers

12.Lack of clinical resources

The absence or limited availability of local resources was raised as a challenge by a range of health care professionals across six studies (21.4%). Resources which were particularly highlighted included insufficient laboratory facilities [50], and a lack of pre- and post-test genetic counselling [28], with only 13% of surveyed psychiatrists aware of professionals offering genetic counselling locally [40]. The need for sufficiently skilled coordinating staff to oversee implementation was also raised as a limiting factor [26]. In line with the challenging financial costs associated with implementation discussed above (Section 3.4.1), there were also concerns reported with regards to budget reallocation [27], particularly away from psychosocial therapies [37].

13.Lack of effective interventions

Furthermore, the lack of appropriate and effective interventions or treatments in the case of a high-risk outcomes or positive genetic test was identified across several studies (*n* = 3; 10.7%) [26,46,49]—in fact, one study reported a substantial decline in interest in hypothetical genetic testing when corresponding treatments were unavailable [46].

##### Facilitators

14.Availability of associated infrastructure

Several studies (*n* = 4; 14.3%) reported the need for precision models, particularly genetic testing and pharmacogenomics, to be offered directly through specialist providers [35,40,42]. Clinicians further highlighted the need for wider availability of testing and related infrastructure to optimize implementation [33].

15.Integration into current workflow

The seamless integration into current workflow was also highlighted as a facilitator within three studies (10.7%) [29,38], including suggestions of improved integration of genotyping into primary care and general health check-ups [49].

16.Availability of effective interventions and counselling

Finally, health care professionals and service users alike highlighted the local availability of subsequent, appropriate intervention in the event of a high-risk outcome [31] and pre- and post-test counselling provisions for genetic testing as facilitating factors [28] across two studies (7.1%).

#### 3.4.3. Outer Setting

One quarter of the included literature (25.0%; *n* = 7) identified barriers or facilitators relating to features of the outer setting which may impact upon implementation.

##### Barriers

17.Potential misuse of personal data

The barriers identified in relation to the outer setting were largely associated with safety and security concerns (*n* = 4; 14.3%) regarding the potential for the misuse of personal data by external stakeholders and third parties. Such challenges were most commonly reported by service users as opposed to other stakeholder groups. A number of genetic-based studies reported the misuse of private genetic data by third-parties as a potential barrier, particularly if the data were available to employers or insurance companies [43,45,47,49]. This was specifically highlighted for direct-to-consumer genetic testing [49].

18.Ethics of risk communication

Additionally, general ethical concerns were also reported in one study (3.6%) with regards to using risk communication tools as a ‘persuasive mechanism’ to influence decision-making [48].

##### Facilitators

19.Confidentiality of personal data

Health care professionals and service users alike emphasized the need for the confidentiality of genetic data and the outcome of genetic tests across two studies (7.1%) [28,43].

20.Compliance with law and regulatory pathways

Finally, continued compliance with applicable law, regulations and safety planning was also acknowledged as a factor in a single study (3.6%) to improve the ethical implementation of a prediction model [26].

#### 3.4.4. Characteristics of the Individuals—Staff

Characteristics of health and social care staff were highly reported as potential barriers and facilitators to the implementation of precision psychiatry methods, with 62.07% (*n* = 18) of the included literature reporting at least one factor relating to this construct.

##### Barriers

21.Negative staff perceptions of precision medicine

Negative staff perceptions of the usefulness and applicability of precision psychiatry were commonly reported (*n* = 6; 21.4%), particularly with regards to genetic testing and genomics. Across the literature, there was a general sense that genomics in psychiatry was still in its infancy and as such, testing was not currently clinically useful [28], with other professionals reporting that tests were not ready for routine implementation and were only currently applicable in academic research settings [33]. Limiting factors to the perceived usefulness of such testing were also reported; whilst health care professionals reported interest in hypothetical tests with high predictive power, this interest dropped substantially for tests with moderate prognostic/predictive power [35]. Doctors generally expressed more concern about the accuracy of pharmacogenomic testing compared with pharmacists [33], and other professionals perceived testing as only useful under specific circumstances (e.g., treatment resistance) [33]. Furthermore, clinicians also expressed concern of the potential impact of precision models upon the staff-service user dynamic, citing that precision psychiatry may offer less personal and empathic care compared with current clinical care [34]. Several studies also highlighted important discrepancies in attitudes across different clinical professions. For example, clinical scientists held significantly more favourable attitudes towards pharmacogenomic testing compared with core psychiatrists [30], and professionals who were less comfortable with prescribing pharmacological interventions for alcohol use disorders acknowledged that this would limit the likelihood of using genetic testing to guide treatment decisions [50].

22.Poor perceived competence in precision medicine

Poor perceived competence and abilities of health care professionals was frequently reported as a barrier (*n* = 7; 25.0%), most commonly in relation to genomics and genetic testing. A survey of health care professionals highlighted general poor perceived competency in pharmacogenomics, with pharmacists generally reporting higher competence than doctors across various aspects of pharmacogenomics [33]. Several further studies reported low confidence in skills and knowledge surrounding genetic testing and feelings of being ill-equipped or inadequately trained to discuss genetic information [37,42,45,51], with just 9% of respondent clinicians feeling competent to offer genetic testing and interpret the results [40], thus raising the possibility that a more formal specialized clinical service is required [42]. However, two further studies reported that the majority of medical geneticists and genetic counsellors surveyed had not received any training in psychiatric genetics [51] and there was a lack of familiarity of such tests among laboratory staff [27], suggesting this is a system-wide issue across various environments and professionalisms.

23.Poor previous experience

Poor previous experience with precision medicine methods was also reported as an obstacle within a single study (3.6%) and was reported to result in more widespread negative perceptions of precision medicine methods; for example, several clinicians voiced skepticism over the accuracy of a clinical decision support tool based largely on poor previous experience with similar systems [38].

24.Lack of motivation to address mental health in primary care

Within a single study (3.6%), primary care professionals emphasized a lack of motivation to address mental health problems within primary care settings, resulting in poorer engagement with the implementation process in this environment [29].

##### Facilitators

25.Adequate skills and competence training

The reported facilitators across this construct related almost entirely to competent, knowledgeable, and skilled staff in combination with the corresponding availability of education and training; this was also the most highly reported overall facilitator (*n* = 7; 25.0%). In particular, facilitators included staff who were competent in communication and in interpreting the results of the precision model [28,29,31]. Health care professionals highlighted the need for more extensive and continuous training in communication skills [29,31], interpreting the outputs of a precision model [28,29], statistical methods [45], genetics and familial risks [42], and case-specific training and feedback [24]. One study identified that improved competence and knowledge in pharmacogenomics led to improved confidence amongst psychiatrists [44], whilst another study reported that accrued clinician experience in the implementation of a risk calculator improved skills [29].

26.Effective time management and organization

Finally, effective schedule management and organization of physicians was identified (*n* = 1; 3.6%) as a mitigating factor against the potential time investment of implementing a new precision model [29].

#### 3.4.5. Characteristics of the Individuals—Service Users and Families/Caregivers

Similarly, the characteristics and needs of the potential service users (and families/caregivers) were also highly reported as barriers or facilitators to the implementation of precision psychiatry, with 64.3% (*n* = 18) reporting at least one factor within this construct.

##### Barriers

27.Potential psychological harm

The most commonly reported barrier within this construct (*n* = 11; 39.3%), and indeed the most commonly reported overall barrier by a range of stakeholders, was the potential for an adverse psychological impact upon the service user. The potential for psychological harm was widely discussed [28,33,42], including the possible anxiety-inducing effects of risk prediction [27,29,39,41,45], the risk for fatalistic thinking or an exacerbation of depressive symptoms following a high-risk outcome [48,49], and subsequent decreased motivation [50].

28.Potential economic and occupational harm

Further adverse impacts upon the service user and their families were also widely considered, including the potential for economic and occupational harm (*n* = 8, 28.6%). Challenges included the potential for general economic harm [33] and the denial of insurance [28,37,47,51], concerns over privacy [45,49], potential genetic discrimination and inadequate legal support for such discrimination [45,49], and employment discrimination [28,37,41].

29.Potential stigmatization

Similarly, adverse social effects were also raised by several studies (*n* = 4; 14.3%), such as possible stigmatization [42,49], potentially unwarranted concerns for offspring [47], and lowered expectations for children carrying a high-risk gene [37].

30.Skepticism regarding genetics

There was also a sense of skepticism of psychiatric genetics across the literature (*n* = 2; 7.1%); those who believed that mood disorders were predominantly caused by genes were significantly more interested in genetic testing than those who placed more emphasis on the causative nature of life experiences [35], with some participants opposing genetic testing completely [43], and concerns that prenatal genetic testing would impact upon decisions surrounding pregnancy [35].

31.Negative attitudes towards psychiatry

One study (3.6%) identified general negative attitudes towards psychiatry and a deep mistrust in the psychiatric profession amongst service users as a challenging barrier to overcome [31].

32.Resistance to knowing risk scores

A resistance to knowing risk scores for major depression was raised as a barrier amongst service users within one study (3.6%), particularly compared to physical health conditions; crucially, this was discussed in light of the lack of definitive preventive treatment available for depression [31].

33.Fear of invasive procedures

Just one study (3.6%) identified specific phobias relating to precision methods as a challenging factor, such as the fear of needles involved in genetic testing [24].

34.Weak demand and engagement

Further negative perceptions and attitudes towards services were raised as barriers in one study (3.6%), including weak service user demand for genetic psychiatry services, poor service user engagement with the service and negative past experiences with similar services [42].

##### Facilitators

35.Service user engagement

Patient engagement was highlighted as a facilitating implementation factor in one study (3.6%), with a potential positive impact reported upon compliance, motivation and adherence to treatment [50].

36.Trusting service user/clinician relationship

A further study (*n* = 1; 3.6%) also reported that a trusting relationship between the health care professional and the service user is essential, and was posited to improve compliance [39].

#### 3.4.6. Implementation Process

Just 10.7% (*n* = 3) of the included literature reported factors relating to the actual process of implementation. No barriers were reported in relation to the implementation process, but this was secondary to the dearth of actual implementation studies within the psychiatric literature.

##### Facilitators

37.Outreach to local clinicians and clinical prompts

Facilitators involved in the implementation process exclusively related to engagement and outreach to local professional and public communities. Two implementation studies (7.1%) reported an improved implementation process when clinician prompts and outreach activities were conducted [25,26]. Further, the content of clinical prompts was also of importance, with a raised response rate reported when the patient names were used instead of internal identification number usage [25].

38.Engagement and community outreach

Finally, the critical importance of service user and public involvement initiatives and public education surrounding the use of genomics was highlighted in a single study (3.6%) [36].

## 4. Discussion

This large-scale review demonstrates several key strengths; we implemented a wide-reaching and inclusive search strategy to ensure a variety of literature from various academic fields was captured. Furthermore, no limitations were placed on date of publication, geographical location and manuscript language, ensuring we identified research from a variety of precision approaches and geographies. To our knowledge, this is also the first systematic review to identify factors affecting the implementation of global precision methods in psychiatry. These factors represent a framework of key considerations to address during both the development and implementation of precision prediction models in psychiatry and as such, will be of use to a wide range of professionals, from research scientists to health care professionals and policymakers alike. In particular, a number of key findings may help to shape protocols for future implementation science.

### 4.1. Key Findings

The current literature identified a range of barriers, and a narrower selection of facilitators, highlighting the numerous and widespread challenges which may be encountered during the implementation process. The most highly reported barriers covered a range of issues, such as ethical considerations, logistical challenges, and attitudinal perceptions of key stakeholders, thus demonstrating that there is no singular challenging factor to overcome for successful implementation. In particular, we identified five barriers which were most commonly reported across the literature: (i) Potential psychological harm (*n* = 11), (ii) Cost and time investments (*n* = 9), (iii) Potential economic and occupational harm (*n* = 8), (iv) Poor accuracy and utility of the model (*n* = 8), (v) Poor perceived competence in precision medicine (*n* = 7), and one facilitator: (i) Adequate skills and competence training (*n* = 7).

First, the potential adverse effect of precision methods upon the service users’ psychological wellbeing was identified across much of the literature. The potential negative psychological effects of receiving a high-risk outcome received particular consideration due to the potential for fatalistic thinking and fear of the future, alongside the subsequent potential social implications such as stigmatization. This is a particularly pertinent consideration given the lack of definitive testing currently available in psychiatry. This topic has received much debate across psychiatry, law and bioethics—particularly in relation to psychosis risk [52,53]. Although qualitative research suggests that such risk labels may have an impact upon the sense of self and social stigma [54,55,56,57], these adverse effects are lessened when participants are fully informed of the meaning of ‘at-risk’ [57,58]. This highlights the crucial importance of training staff to interpret and discuss the output of risk prediction models, and appropriately feed this information into the clinical decision-making process. Moreover, qualitative explorations with young people at-risk have actually highlighted the therapeutic effects of sharing their experiences within specialist services [59,60]. Indeed, there is a wide range of vocational, psychosocial and familial support interventions which are offered through such specialist services which might not otherwise be made available to the individual [61]. Further recent research has also disputed the adverse social impact of a psychosis-risk label [58], and recent evidence suggests that the presence of stigma is actually associated with the service users’ experiences of distressing symptoms as opposed to the clinician’s designation [54].

Second, this review also identified various logistical challenges, particularly surrounding the high financial cost of implementing precision models compared to current psychiatric care. This finding necessitates a greater research focus on economic modelling of such approaches, particularly high-cost methodologies such as magnetic resonance imaging, in order to determine cost-effectiveness. Similarly, a range of health care professionals highlighted the facilitating nature of precision models which are quick and easy to use in practice, suggesting that this aspect needs to be given greater weight in model development rather than building complex, multi-modal models which may be more challenging to implement in practice. However, in contrast, one included study identified the integration of multi-modal biological data as a facilitating factor [39]. This suggests that model development requires a trade-off to find balance between model simplicity and predictive accuracy. Ultimately, these findings reinforce the notion that precision psychiatry models should be developed with such logistical challenges in mind.

Third, numerous included studies also discussed the potential adverse financial and occupational impact of precision models in relation to insurability and employment of the service user, although, a significant proportion of the studies reporting this barrier were derived from research undertaken in the United States [28,37,41,47], and so this challenge may require more substantial consideration within countries adopting an insurance-based health care system. Nevertheless, the frequent reporting of these concerns may also reflect widespread mistrust in personal data security and emphasizes the need to place priority on the safety and security of data, particularly as data-driven health care continues to build at pace, globally. Taken together with the highly-reported concerns regarding psychological wellbeing discussed above, it will be imperative within ongoing research efforts to develop an ethical framework for the implementation of preventive psychiatry approaches in close collaboration with the key stakeholder groups outlined in this review—with a particular focus on service users and their families to ensure further research ‘optimizes benefit and minimizes harm’ [61].

Fourth, challenges regarding poor clinical utility and accuracy were highly reported. Indeed, our recent review [6] reported highly variable rates of prognostic/predictive accuracy across precision psychiatry models within the literature, and further identified a high risk of bias across 94% of the models (according to the PROBAST [62] tool). Moreover, due to a lack of external validation efforts present across much of the literature [6,63], it is difficult to assess the real-world utility of many existing clinical prediction models. Thus, in order to progress in this field, a shift towards further external validation and implementation is needed. Recent studies have started focusing on the external replication of existing clinical prediction models in the field of psychosis risk, and should be followed by similar initiatives in other fields [64].

Finally, we identified a substantial proportion of records reporting poor perceived competence and knowledge amongst staff in precision medicine, with a particular focus on genetic testing and pharmacogenomics. Whilst there is evidence of substantial progress in genomic education amongst medical professionals in recent years [65,66], this current review suggests that a more tailored approach may be required within psychiatry. In parallel, it is unsurprising that the most highly reported facilitator across the literature was the immediate need for greater provision of training and education amongst staff, especially in relation to genetics and pharmacogenomics. This was similarly reflected within a recently developed implementation plan for general precision medicine informed by stakeholder views [17] and reiterates the importance of frequent adjustments to national medical training and teaching curricula to reflect recent scientific advances, as recently argued by our group [2].

Overall, these key findings provide a tentative empirical platform to guide future implementation research and ensure that model development and validation activities take implementation into account from the very early stages. It is commonplace within research that implementation barriers are considered too late, when the model is already developed and validated, leading to the frequent development of tools which are impractical to implement in practice. As such, these current findings will be of interest to research scientists as well as policymakers who must ensure that implementation activities are sufficiently regulated by a comprehensive legal framework. Finally, this research may be of importance to the end-users of such innovation: health care professionals and service-users.

### 4.2. Limitations and Future Directions

Whilst this review covered a comprehensive range of literature, it also highlighted numerous current gaps in the literature base which necessitate further research attention. No studies were identified which assessed the perspectives of policymakers; in light of their integral role in the provision and regulation of health care, it will be important to gather their perspectives in future research. Furthermore, the majority of our results were based upon qualitative literature. However, the nuance and detail which arises from qualitative research is essential to move forward in this field and the qualitative synthesis presented here was best suited to the available literature.

Finally, this review also reiterated the sparsity of implementation studies for precision methods in the field of psychiatry, previously acknowledged in a recent review [6]. As such, ‘Process’ factors were limited within the included literature, with no barriers reported for this CFIR construct. This highlights an important gap in the research and necessitates a focus on translational research fit for implementation in order to establish a high-quality evidence base in support of precision psychiatry. Whilst a RCT was recently published which assessed the clinical effectiveness of a predictive algorithm to guide antidepressant treatment [67], it does not meet inclusion criteria for this review. However, qualitative experiences of this trial are due to be reported separately and may offer higher-grade evidence of the barriers and facilitators to real-world implementation of precision psychiatry. Meanwhile, at present there are a variety of established implementation frameworks and guidelines, beyond the CFIR [23], which can support implementation efforts across the field, such as the Normalization Process Theoretical Framework [68], the FAIR principles [69], the RE-AIM framework [70], and the Expert Recommendations for Implementing Change (ERIC) project [71], as well as existing systematic reviews regarding other related fields of innovation in psychiatry, such as digital and eMental Health [72,73].

## 5. Conclusions

A diverse range of barriers exist which may impede the translation of precision psychiatry research findings into real-world clinical care; these barriers should be taken into consideration during the early development stages of such models and corresponding pragmatic and innovative solutions are required at the level of the individual, the organization and the wider system. Ongoing community engagement initiatives are essential to address negative perceptions amongst stakeholders, and specialized medical educational modules in precision methods are required to equip health care staff with the knowledge and skills needed to confidently employ precision methods in practice. Finally, this review has highlighted the vital need for further implementation research, education and training. In particular, future research should prioritize the initiation of randomized controlled trials of precision models in order to successfully translate precision psychiatry research from the ‘bench to bedside’.

## Figures and Tables

**Figure 1 brainsci-12-00934-f001:**
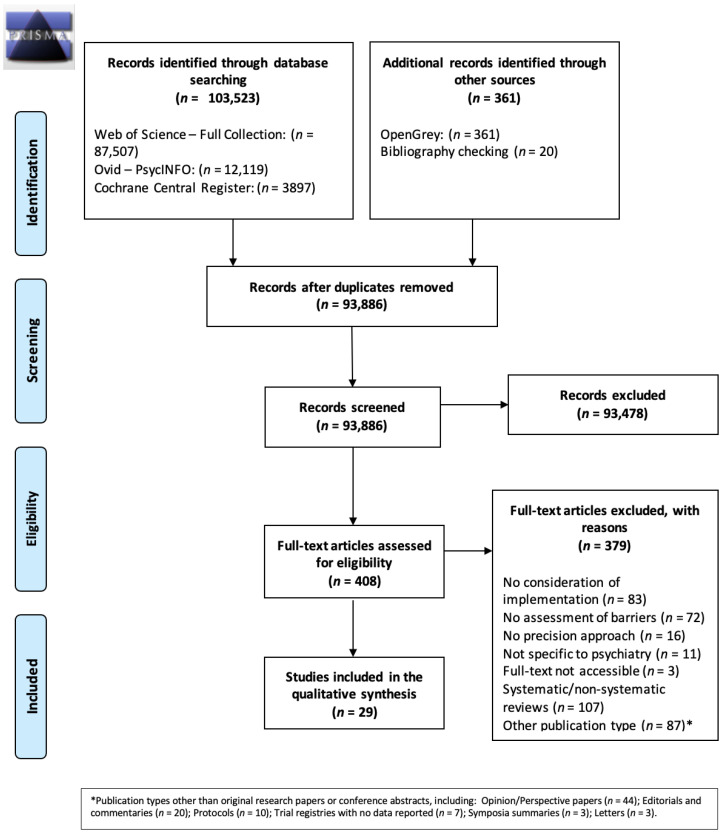
Preferred reporting items for systematic reviews and meta-analyses (PRISMA) diagram detailing the full study selection process.

**Figure 2 brainsci-12-00934-f002:**
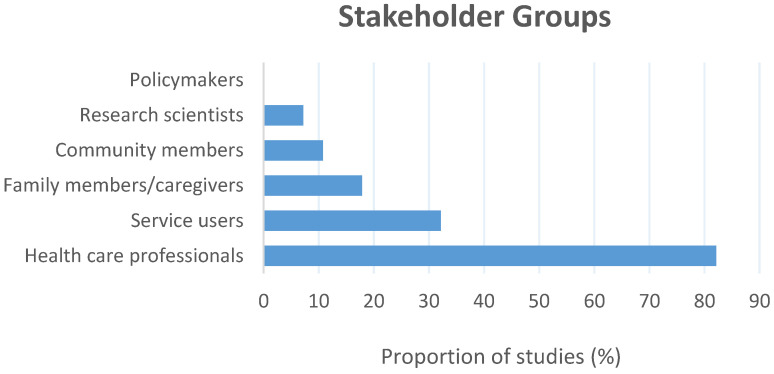
The stakeholder groups investigated across the included literature and the proportion (%) of studies consulting each group.

**Figure 3 brainsci-12-00934-f003:**
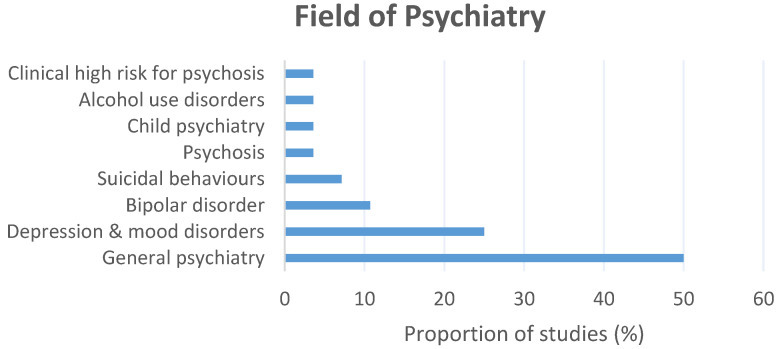
The fields of psychiatry investigated across the included literature and the proportion (%) of studies investigating each field.

**Figure 4 brainsci-12-00934-f004:**
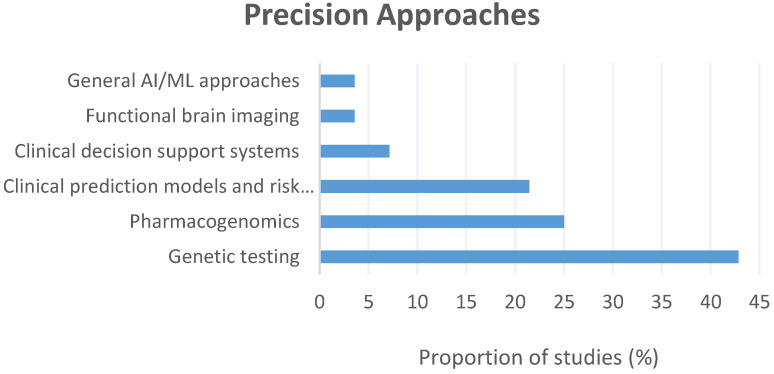
The precision psychiatry approaches adopted across the included literature and the proportion (%) of studies investigating each approach.

**Figure 5 brainsci-12-00934-f005:**
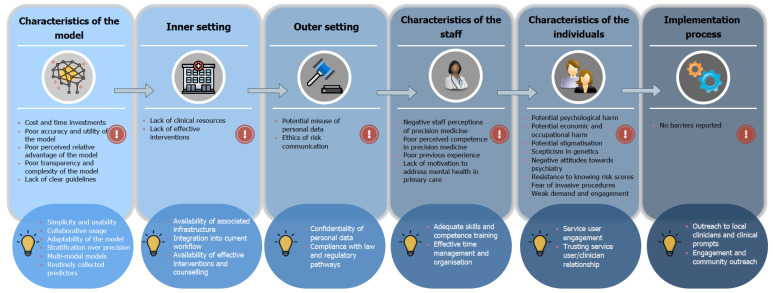
A high-level visual summary of the identified barriers and facilitators which may impact upon the real-world implementation of precision psychiatry approaches, structured according to the Consolidated Framework for Implementation Research (CFIR).

**Figure 6 brainsci-12-00934-f006:**
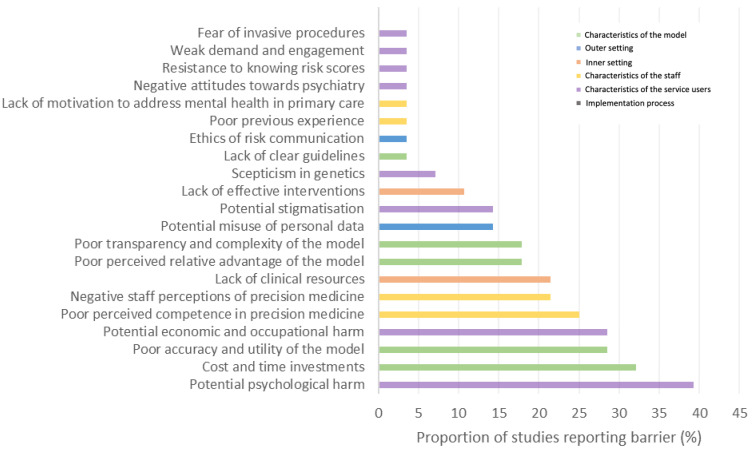
Proportion (%) of the included studies reporting each barrier.

**Figure 7 brainsci-12-00934-f007:**
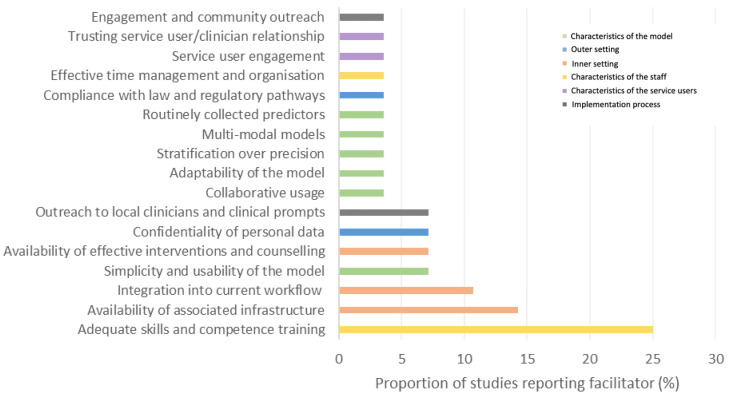
Proportion (%) of the included studies reporting each facilitator.

**Table 1 brainsci-12-00934-t001:** Inclusion and exclusion criteria.

	Inclusion Criteria	Exclusion Criteria
(I) Consultation with key stakeholders	▪ Health and social care professionals ▪ Service users and family members and/or caregivers ▪ Policymakers ▪ Research scientists ▪ Local community members	NA
(II) Precision psychiatry approach	▪ Diagnostic, predictive or prognostic models employing a precision (or stratified) approach ▪ Specific to the field of psychiatry (i.e., any DSM/ICD diagnosis)	▪ Precision models relating to traumatic brain injury, neurological disorders or dementias
(III) Predictors	▪ Clinical ▪ Sociodemographic ▪ Service use ▪ Behavioural ▪ Biomarkers (neuroimaging, genomic, pharmacogenomic, metabolomic, cognitive) ▪ Any combination of the above	NA
(IV) Study design	▪ Primary research studies which consider actual or proposed implementation ▪ Qualitative, quantitative or mixed methods	^▪^ Secondary research (systematic and non-systematic reviews, and meta-analyses) ^a^
(V) Assessment of barriers/ facilitators	▪ Systematic assessment of barriers and/or facilitators to precision (or stratified) psychiatry	▪ Assessment of barriers and/or facilitators only raised in the discussion section
(VI) Publication type	▪ Conference abstracts ^b^ ▪ Full journal articles	▪ Protocols ▪ Editorials, letters and commentaries ▪ Expert opinion papers ^c^
(VII) Level and quality of evidence	▪ Any level or quality of evidence	NA
(VIII) Language	▪ Any language	NA
(IX) Publication date	▪ Published from database inception to 25 October 2020	NA

^a^ Reviews were screened for relevant research via the hand-searching of bibliographies. ^b^ Conference abstracts were only included if they fit all other criteria, including the reporting of primary research data. ^c^ Expert opinion papers were flagged for inclusion should the final literature base be too narrow to facilitate sufficient discussion (<5 studies). DSM = Diagnostic Statistical Manual (any version); ICD = International Classification of Diseases (any version).

**Table 2 brainsci-12-00934-t002:** Characteristics of the included studies.

First Author, Date	Location	Research Method	Field of Psychiatry	Type of Precision Model	Sample	Level of Evidence	Summary of Barriers	Summary of Facilitators
Bellón, 2014 [31]	Spain	Focus groups	Depression	Individualised risk prediction algorithm	52 service-users	1	Resistance to knowledge of risk scores; Negative attitudes towards psychiatry	Adequate skills and competence training; Availability of effective interventions and counselling
Brown, 2020 [32]	United States	Survey	Suicidal behaviours	Individualised risk prediction algorithm	139 health care professionals (psychologists, social workers, psychiatrists, nurses and other allied health professionals)	1	Poor accuracy and utility of the model; Poor transparency and complexity of the model	N/A
Chan, 2017 [33]	Singapore	Survey	General psychiatry	Pharmaco-genomics	167 doctors and 27 pharmacists (*n* = 194)	1	Poor perceived competence in precision medicine; Negative staff perceptions of precision medicine; Cost and time investments; Lack of clear guidelines; Potential psychological harm; Potential economic and occupational harm	Availability of associated infrastructure
Doraiswamy, 2020 [34]	North America, South America, Europe and Asia-Pacific	Survey	General psychiatry	General AI/ML applications	791 psychiatrists	1	Poor perceived relative advantage of the model; Negative staff perceptions of precision medicine	N/A
Dunbar, 2012 [27] ^†^	New Zealand	Surveys and interviews	General psychiatry	Pharmaco-genomics	33 senior medical officers and registrars	1	Lack of clinical resources; Poor perceived competence in precision medicine; Poor perceived relative advantage of the model; Cost and time investments; Potential psychological harm	N/A
Erickson, 2013 [35]	United States	Survey	Mood disorders	Genetic testing	147 service users, caregivers and community members, and mental health professionals	1	Negative staff perceptions of precision medicine; Scepticism in genetics	Availability of associated infrastructure
Evanoff, 2016 [36] *	United States	Stakeholder meetings	General psychiatry	Genomics	Health care professionals, research scientists, and community members (*n* = unspecified)	1	Poor accuracy and utility of the model	Engagement and community outreach
Finn, 2005 [37]	United States	Survey	General psychiatry	Genetic testing	844 psychiatrists or psychiatrists-in-training	1	Potential stigmatisation; Potential economic and occupational harm; Lack of clinical resources; Poor perceived competence in precision medicine	N/A
Goodspeed, 2019 [38]	United States	Focus groups and prototype development	General psychiatry	Pharmaco-genomics integrated into a clinical decision support system	16 mental health clinicians	1	Poor perceived relative advantage of the model; Poor previous experience; Cost and time investments	Integration into current workflow
Henshall, 2017 [39]	United Kingdom	Focus groups and prototype feedback	General psychiatry	Clinical decision support system	12 consultant psychiatrists, 11 primary care practitioners and 8 patients/carers (*n* = 31)	1	Poor perceived relative advantage of the model; Potential psychological harm; Poor transparency and complexity of the model; Cost and time investments; Poor accuracy and utility of the model	Collaborative usage; Trusting service user/clinician relationship; Multi-modal models; Simplicity and usability of the model
Hoop, 2008 [40]	United States	Survey	General psychiatry	Genetic testing	45 psychiatrists	1	Lack of clinical resources; Poor perceived competence in precision medicine	Availability of associated infrastructure
Hoop, 2010 [28] ^†^	United States	Survey	General psychiatry	Pharmaco-genomics	75 psychiatry attending physicians and residents	1	Potential psychological harm; Potential economic and occupational harm; Poor accuracy and utility of the model; Cost and time investments; Lack of clinical resources; Negative staff perceptions of precision medicine	Confidentiality of personal data; Adequate skills and competence training; Availability of effective interventions and counselling
Illes, 2008 [41]	United States	Survey	Major depression	Functional brain imaging	52 psychiatrists or psychologists, and 72 inpatient and outpatient service users (*n* = 124)	1	Cost and time investments; Potential psychological harm; Potential economic and occupational harm	N/A
Jenkins, 2016 [42]	United Kingdom	Face-to-face and telephone interviews	General psychiatry	Genetic testing	9 psychiatric staff nurses and consultant psychiatrists	1	Poor transparency and complexity of the model; Poor accuracy and utility of the model; Poor perceived competence in precision medicine; Weak demand and engagement; Potential psychological harm; Potential stigmatisation	Availability of associated infrastructure; Adequate skills and competence training
Laegsgaard, 2008 [43]	Denmark	Questionnaire	General psychiatry	Genetic testing	681 patients and relatives	1	Potential misuse of personal data; Scepticism in genetics	Confidentiality of personal data
Lucero, 2020 [44] *	United States	Survey	General psychiatry	Pharmaco-genomics	830 psychiatrists	1	N/A	Adequate skills and competence training
Mathews, 2018 [24] *^,^^†^	United States	Feasibility study	Child psychiatry	Pharmaco-genomics	Parents and associated clinicians of 73 young service users	2	Cost and time investments; Fear of invasive procedures	Adequate skills and competence training
Moreno-Peral, 2018 [29] ^†^	Spain	Face-to-face semi-structured interviews	Major depression	Individualised risk prediction algorithm	67 family physicians	2	Cost and time investments; Poor transparency and complexity of the model; Lack of motivation to address mental health in primary care; Potential psychological harm	Simplicity and usability of the model; Stratification over precision; Integration into current workflow; Adequate skills and competence training; Effective time management and organisation
Oliver, 2020 [25] ^†^	United Kingdom	Feasibility study	Clinical high-risk for psychosis	Transdiagnostic risk calculator	Clinicians of 3722 patients screened and independent consultation with an unspecified number of service users and clinicians	2	N/A	Routinely collected predictors; Outreach to local clinicians and clinical prompts
Reger, 2019 [26] ^†^	United States	Case example	Suicidal behaviours	Clinical prediction model	A clinical implementation team of professionals	2	Lack of effective interventions; Lack of clinical resources	Outreach to local clinicians and clinical prompts; Compliance with law and regulatory pathways
Salm, 2014 [45]	United States	Survey	General psychiatry	Genetic testing	372 psychiatrists and 163 neurologists (*n* = 535)	1	Potential misuse of personal data; Poor perceived competence in precision medicine; Potential psychological harm; Potential economic and occupational harm	Adequate skills and competence training
Smith, 1996 [46]	United States	Survey	Bipolar disorder	Genetic testing	48 members of a bipolar disorder support group, 35 medical students and 30 psychiatry residents (*n* = 113)	1	Lack of effective interventions	N/A
Trippitelli, 1998 [47]	United States	Questionnaire	Bipolar disorder	Genetic testing	90 service users and their spouses	1	Potential misuse of personal data; Potential stigmatisation; Potential economic and occupational harm	N/A
Wachtler, 2018 [48]	Australia	Focus group, prototype development and semi-structured interviews	Depression	Clinical prediction model	17 members and of the community and 7 service users (*n* = 24)		Poor transparency and complexity of the model;Ethics of risk communication; Potential psychological harm; Poor accuracy and utility of the model	Adaptability of the model
Walden, 2015 [30] ^†^	Canada	Survey	General psychiatry	Pharmaco-genomics	168 physicians who had ordered pharmaco-genomic tests for psychotropic medication	1	Negative staff perceptions of precision medicine	N/A
Wilde, 2010 [49]	Australia	Focus groups	Major depression	Genetic testing	36 members of the public (14 with disclosure of family history of mental illness)	1	Poor perceived relative advantage of the model; Poor accuracy and utility of the model; Potential misuse of personal data; Lack of effective interventions; Potential psychological harm; Potential stigmatisation; Potential economic and occupational harm	Integration into workflow
Williams, 2016 [50]	United States	Semi-structured interviews	Alcohol use disorders	Genetic testing	24 primary care providers	1	Cost and time investments; Poor accuracy and utility of the model; Lack of clinical resources; Negative staff perceptions of precision medicine; Potential psychological harm	Patient engagement
Zhou, 2014 [51]	Australia	Survey	Schizophrenia, bipolar disorder and major depression.	Genetic testing	104 psychiatrists, 36 genetic counsellors, and 17 medical geneticists (*n* = 157)	1	Poor perceived competence in precision medicine; Potential economic and occupational harm	N/A

Records marked with * represent a conference abstract; all other records are full journal publications. Records marked with **^†^** represent those employing actual implementation methods as opposed to hypothetical or simulated implementation. Quality ratings: 3 = Randomized Controlled Trials, 2 = Pilot and feasibility studies, 1 = All other primary research involving stakeholder consultation. AI = Artificial Intelligence; CDS = Clinical Decision Support; ML = Machine Learning.

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
