# Peer review of "Real-World Implementation of Precision Psychiatry: A Systematic Review of Barriers and Facilitators"

_brainsci, 2022, doi:10.3390/brainsci12070934_

Round 1

Reviewer 1 Report

Thank you for a comprehensive, well presented literature review highlighting the barriers and facilitators to precision psychiatry implementation. The introduction provided a range of relevant background information with well selected supporting literature. The methods were appropriate, relevant and clearly described. The inclusion and exclusion criteria are valid and aligned with a robust search strategy, although 93000+ is a very large number for abstract screening. In particular, the data analysis section was very well described. 

The results are well presented and the data shows a good range of clinical fields, stakeholder groups and approaches to precision. The visual aids (graphs and tables) were clear and relevant to the research question. 

The discussion demonstrates high level, relevant thinking in terms of implementation and real-world applicability. There is good alignment to literature. I would refrain from including results in this section ( numbers of studies reporting on barriers). The limitations are well defined. The conclusions and discussion points align well with the overall aim of the literature review and provide a clear platform to guide future implementation research.  A very well done review. 

Author Response

Thank you so much for your kind feedback on the manuscript, it is much appreciated. Please see the attached cover letter for a detailed discussion of the minor revisions made to the manuscript. 

Reviewer 2 Report

This is a very interesting paper based on the implemention of precision psychiatry in real-world. The authors conducted a systematic review and meta-analysis with the aim to identifying barriers and facilitators affecting the implemention of precision psychiatry in clinical care. The paper is well-written, and is really of interest. However, I would propose several minor changes before considering it for publication.

How many papers were initially retrieved in the search? In the abstract the authors report that 28 were included. I consider important to say 28 from a total number of X records. The readers should understand the magnitude of the evidence studied.

In the introduction section the authors should briefly clarify differences between personalized and precision medicine.

The main aims of the study should be expanded and placed in a subsection of the introduction.

The methods are adequately described and the systematic review has been registered in PROSPERO. With respect to Table 1, some clarifications are needed. In the publication type criteria for inclusion, the authors report that conference abstracts and full-journal articles were potentially included, byt letters and editorials were excluded. Could the authors specify if it depended on the study design that was presented? For instance, a conference paper can be about expert opinion data, or based on original data. The same for letters, and editorials...

What dabatases were used? They should briefly described in the abstract section.

Of the 28 included studies, 16 reported barriers or facilitators related to characteristics of the precision model. Should it be interpreted that more translational research is necessary to be conducted? Are precision models to be better designed in terms of their applicability in real-world?

The description of barriers and facilitators in the results section should be numbered. 

Conclusions are brief. I would expand it focusing on barriers and facilitators, and briefly reporting the major future directions for further studies.

Author Response

Thank you so much for your positive feedback on the manuscript, it is much appreciated. Thanks also for your constructive feedback - we have considered each of your points and have made a corresponding edit to incorporate each of your considerations. Please see attached a detailed address of each of your comments. 
